# MEMORY PROXY MAPS FOR VISUAL NAVIGATION

## ABSTRACT

Visual navigation takes inspiration from humans, who navigate in previously un-
seen environments using vision without detailed environment maps. Inspired by
this, we introduce a novel no-RL, no-graph, no-odometry approach to visual nav-
igation using feudal learning to build a three tiered agent. Key to our approach is
a *memory proxy map* (MPM), an intermediate representation of the environment
learned in a self-supervised manner by the high-level manager agent that serves
as a simplified memory, approximating what the agent has seen. We demonstrate
that recording observations in this learned latent space is an effective and efficient
memory proxy that can remove the need for graphs and odometry in visual nav-
igation tasks. For the mid-level manager agent, we develop a *waypoint network*
(WayNet) that outputs intermediate subgoals, or waypoints, imitating human way-
point selection during local navigation. For the low-level worker agent, we learn
a classifier over a discrete action space that avoids local obstacles and moves the
agent towards the WayNet waypoint. The resulting feudal navigation network of-
fers a novel approach with no RL, no graph, no odometry, and no metric map; all
while achieving SOTA results on the image goal navigation task.

## 1 INTRODUCTION

Visual navigation is motivated by the idea that humans likely navigate without ever building detailed
3D maps of their environment. In psychology, the concept of cognitive maps and graphs Tolman
(1948); Chrastil & Warren (2014); Peer et al. (2021); Epstein et al. (2017) formalizes this intuition,
and experiments have shown the validity of the idea that humans build approximate graphs of their
environment, encoding relative distances between landmarks. In vision and robotics, these ideas
have translated to the construction of topological graphs and latent maps based primarily on visual
observations. These *visual navigation* paradigms seek new representations of environments that
are rich with semantic information, easy to dynamically update, and can be constructed faster and
more compactly than full 3D metric maps Gupta et al. (2017); Savinov et al. (2018); Chaplot et al.
(2020b); Mirowski et al. (2018); Chen et al. (2019a); Gervet et al. (2023).

In this work, we focus on visual navigation in environments where odometry information is not
readily available, which limits the efficacy of current SOTA work that assumes access to noise-less
GPS+compass sensors as in popular image-goal navigation challenges Yadav et al. (2023). This lack
of odometry data also limits the efficacy of SLAM based methods that require the camera pose to be
known Kwon et al. (2023); Chaplot et al. (2019), graph based methods that rely on distance to define
edge features Kim et al. (2023), and reinforcement learning (RL) methods which use distance-based
rewards Wijmans et al. (2019). Inspired by NRNS Hahn et al. (2021), which questions the necessity
for RL and simulation to create an effective visual navigation agent, this work leverages feudal
learning and latent maps as a memory proxy to show that it is possible to create a performant visual
navigation agent that uses no odometry, no RL, no training in simulators, no graphs, and no metric
maps.

To do this, we take advantage of a three-tiered, feudal learning network structure and show its ben-
efits under supervised and self-supervised learning paradigms. Feudal learning Vezhnevets et al.
(2017) decomposes a task into sub-components, providing performance advantages that we find
particularly well-suited for visual navigation. The feudal framework identifies *workers* and *man-
agers*, and allows for multiple levels of hierarchy (ie. mid-level and high-level managers). Each of
these entities observes different aspects of the task *and* operates at a different temporal or spatial
scale. For navigation in unseen environments, this dichotomy is ideal to make the overall task more

manageable. The worker-agent can focus on local motion, while manager-agents direct navigation and assess when to move to new regions.

Key to our approach is representing the traversed environment with a learned latent map (instead of a graph) that acts as a sufficient memory proxy during navigation. This *memory proxy map (MPM)* is obtained using self-supervised learning. The high-level manager of our feudal learning agent maintains the MPM as the agent navigates in novel environments, and the MPM's density is used to determine when a region is well-explored, and it's time to move away from the current region. A second key aspect to our approach is a *waypoint network (WayNet)* for the middle-level manager which outputs waypoints (visible sub-goals that act as stepping stones towards a certain goal) for the worker agent to move towards. We train WayNet to imitate human exploration policies in a supervised manner using point-click navigation trajectories from the LAVN dataset Johnson et al. (2024). The intuition is that when humans navigate a simulated environment using point-click teleoperation, they use the skill of choosing a single point in the observation to move toward. For example, the chosen point may be toward the end of a hallway, toward a door, or further into a room. We demonstrate that this skill is easily learnable and generalizes to new environments with zero-shot transfer. Finally, we train a low-level worker to choose actions that avoid obstacles in the environment while following this point-wise navigation supervision. We show SOTA performance on the image-goal navigation task in previously unseen Gibson environments Xia et al. (2018) in Habitat AI Savva et al. (2019) (a simulation environment comprised of scans of real scenes).

Our contributions are fourfold: **1)** A self-supervised memory proxy map (MPM) that enables lean, no-odometry, no-graph, no-RL navigation, **2)** A waypoint network (WayNet) for local navigation through supervised learning of human exploration policies, **3)** A hierarchical navigation framework using agents operating at different spatial scales, and **4)** SOTA performance on the image-goal task in Habitat indoor environments (testing and training on different environments).

## 2 RELATED WORK

**Visual Navigation**     Visual navigation aims to build representations that incorporate the rich information of scenes by injecting image-based learning into traditional mapping and planning navigation frameworks Gupta et al. (2017); Chaplot et al. (2019); Devo et al. (2020); Shah et al. (2021a); Seymour et al. (2021). Early work focused on creating full metric maps of a space using SLAM augmented by images Chaplot et al. (2019; 2020b). While full metric maps can be ideal, especially if the space can be mapped before planning, the representations are computationally complex. Topological graphs and maps can lighten this load and provide image data at nodes and relative distances at edges Savinov et al. (2018); Chen et al. (2019b). While easier to build, these methods require odometry to be readily available and have the potential for large memory requirements, especially if new nodes are added every time an agent takes an action Shah et al. (2021a; 2022); He et al. (2023). One solution to this problem is sparser topological graphs Hahn et al. (2021); Shah et al. (2021b) where visual features of unexplored next-nodes are sometimes predicted or hallucinated He et al. (2023). Some methods go a step further by pairing semantic labels with the graph representation Kim et al. (2023); Chang et al. (2023). However, these methods break down in environments that are sparse or featureless, have many duplicate objects, or contain uncommon objects that may not appear in popular object detection datasets. Another solution is to only use graphs during training to build 2D embedding space representations of environments, i.e. potential fields or functions, that preserve important physical Morin et al. (2023); Ramakrishnan et al. (2022); Bono et al. (2023b), visual Henriques & Vedaldi (2018); Bono et al. (2023a); Ramakrishnan & Nagarajan (2022), or semantic Georgakis et al. (2021); Chaplot et al. (2020a); Al-Halah et al. (2022) relationships between regions in the environment. Our method more closely aligns with this line of work, but we do not use any graph networks or graph inference and instead build our 2D latent map using self-supervised contrastive methods. dditionally, unlike many of the methods above, we do not require the agent to have information about the test environment (odometry) before deployment, do not use RL or graphs, or learn 3D metric maps.

**Feudal Learning**     Feudal learning originated as a reinforcement learning (RL) framework Dayan & Hinton (1992); Vezhnevets et al. (2017). Researchers have explored RL for the image-goal visual navigation task Zhu et al. (2017), most notably using external memory buffers Kumar et al. (2018); Fang et al. (2019); Beeching et al. (2020); Mezghan et al. (2022). However, it still suffers from

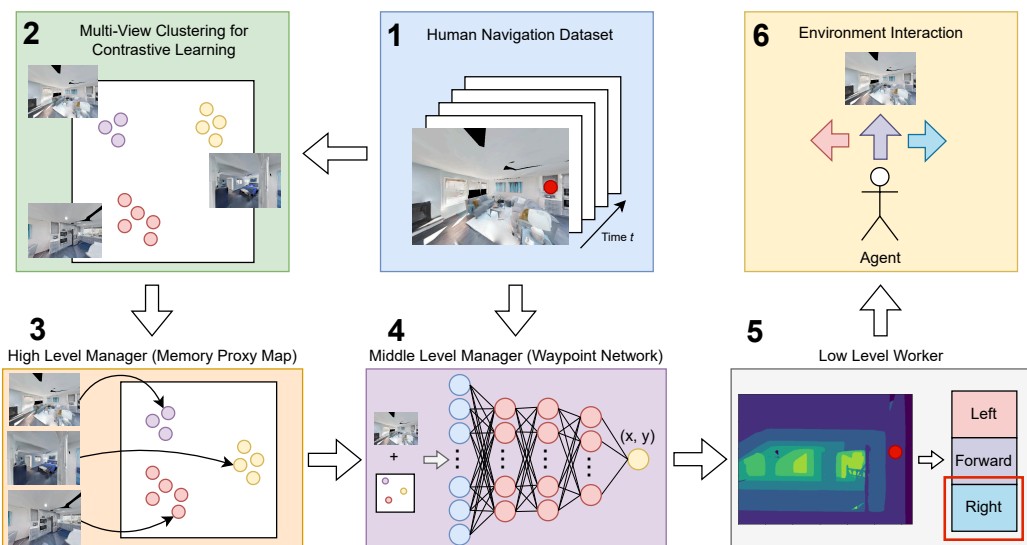

Figure 1: Method Overview **1:** A subset of trajectories of point-click and observation-image pairs are selected from the LAVN dataset Johnson et al. (2024) for learning a latent space for the memory proxy map and training WayNet. We test our method on a separate set of environments. **2:** Images from these pairs are clustered based on feature similarity, and cluster members form positive pairs used for contrastively learning a latent space. **3:** The learned latent space is used to build a memory proxy map where the high level manager (HLM) records a history of agent locations. **4:** The waypoint network (Waynet) is trained to provide subgoals (points) for navigation based on visual observations, imitating human teleoperation via point-clicks. **5:** Based on this point-click guidance and depth map input, the low-level worker predicts to either more forward, left, or right in order to move towards the subgoal (point) and avoid obstacles. **6:** During test time, these low level actions guide agent movement and produce new observations as input for the upper levels of the hierarchy.

several issues such as sample inefficiency, handling sparse rewards, and the long horizon problem Fujimoto & Gu (2021); Le et al. (2018). Feudal reinforcement learning, characterized by its composition of multiple, sequentially stacked agents working in parallel, arose to combat these issues using temporal or spatial abstraction, mostly in simulated environments Vezhnevets et al. (2017) where its effects can be more easily compared to other methods. Some of these task hierarchies are hand defined Vezhnevets et al. (2020), while others are discovered dynamically with Chen et al. (2020) or without Li et al. (2020) human input. The feudal network paradigm has been adopted by other learning schemas outside of RL in recent years, as the hierarchical network structure provides benefits to other methodologies outside of RL. In navigation, hierarchical networks are commonly used to propose waypoints as subgoals during navigation Chane-Sane et al. (2021), typically working in a top-down view of the environment Xu et al. (2021) with only two levels of agents Wöhlke et al. (2021). Our work uses multiple agent levels, operates in the first person point of view for predicting waypoints, and reaps the benefits of this feudal relationship without using reinforcement learning.

## 3 METHODS

**High-Level Manager: Memory** We contrastively learn a latent space that is used to build an aggregate memory proxy map (MPM) that serves as a memory module for our feudal navigation agent. This self-supervised latent space is learned using Synchronous Momentum Grouping (SMoG) Pang et al. (2022) which combines instance level contrastive learning and clustering methods. This model is chosen for its momentum grouping, which enables it to perform both instance-level and group-level contrastive learning simultaneously. We demonstrate empirically that this model improves the quality of the learned latent space comprising the MPM (in Section 6). Instead of using typical data augmentations (e.g. rotation, scale, shift) to determine positive pairs for contrastive learning, we dynamically build clusters of images observed along the training trajectories based on visual similarity

determined by Superglue Sarlin et al. (2020) robust keypoint matching and use these clusters to define positive and negative pairs during training. This step is vital for the success of the agent as using this positive pair definition creates a latent space that preserves the relative distance between images without the need for ground truth odometry data. For each trajectory in the training data, the first image observation is chosen to represent the first cluster center. Each successive image along the trajectory is compared to a memory bank of previously seen cluster centers as the agent travels along its trajectory. If the confidence $\alpha_c$ of these keypoint matches is high, the current image is added to the corresponding cluster. Otherwise, it is used as a new cluster center. We build these clusters per environment for all training trajectories and randomly sample images from the same cluster to act as positive pairs to contrastively train the network.

During inference, observation images are dynamically placed in this contrastively learned latent space to build a memory proxy map (MPM) of previously visited locations on the fly as the agent navigates in novel environments. For greater interpretability, our approach is to build an "isomap imitator network" for a 2D interpretable projection. We train an MLP to map the learned SMoG feature space (128 dim) to a 2D map representation. Specifically, we compute isomap 2D embeddings for the learned SMoG features for all of the training data to reduce the dimensionality of the data while preserving the relative feature distances. Because isomap produces different embedding coordinates each time it is run, a simple MLP network (isomap imitator) is trained to reproduce the isomap embeddings from the SMoG features. To update the MPM during inference, a gaussian weighted circular window with $\sigma = 1$ is added to the corresponding predicted 2D location in the map from the isomap imitator for each image observation, thus creating a density map with values corresponding to the amount of exploration that has occurred in each location. The more an area in an environment is explored, the larger the region in the MPM corresponding to this area, providing valuable information for downstream networks. This allows the agent to roughly localize itself in its environment with respect to its previous observations as well as providing a signal to quantify the amount of exploration that has taken place in a given area. Using this approach allows for adept navigation without the need for graphs or SLAM, resulting in a demonstrably effective and efficient approach via this memory proxy.

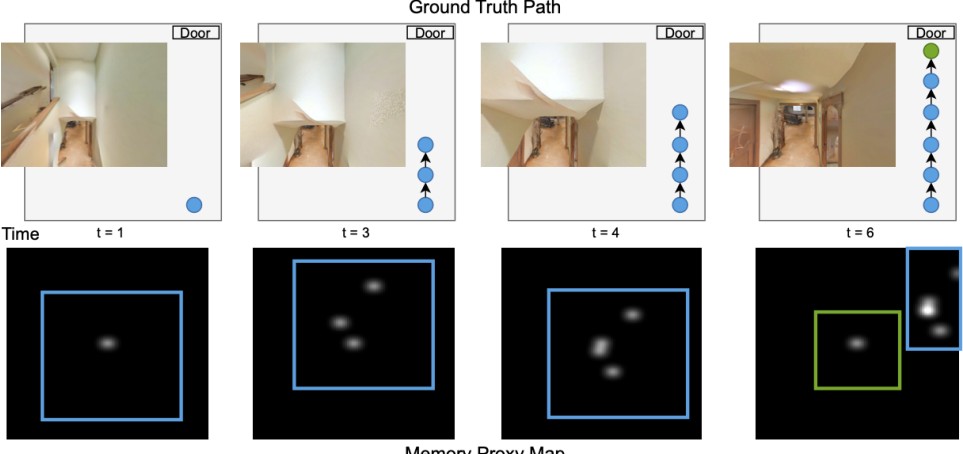

Figure 2: Illustration of the memory proxy map (MPM) during navigation. Row 1: RGB observation images along a trajectory are shown with a diagram of the agent's corresponding location in an environment. The colored circles (blue/green) represent the traveled path. Row 2: The MPM with guassian-weighted occupancy markers corresponding to each observation image. The map is local, of fixed size, and cropped around the most recently added latent map position. In this manner, the agent marks locations in the latent space (not a metric space) corresponding to recently viewed images, thus remembering when observations repeat. Similar observation images cluster together (in blue) until the next view is significantly different in appearance and a new group begins (in green). The MPM is a convenient no-graph mechanism to remember previously visited regions, effective and efficient in the image-goal navigation task to quantify the amount of exploration in a given area of the environment.

**Mid-Level Manager: Direction**   For the mid-level manager, we leverage human knowledge to learn optimal navigation and exploration policies directly from demonstrations collected in a human navigation dataset. The intuition is that the human point-click navigation decisions in the LAVN dataset Johnson et al. (2024) are learnable and generalize to new environments with zero-shot transfer. For each trajectory in LAVN, we use the HLM to create a memory proxy map. Then, using the RGBD observations from the dataset and the corresponding crops of the MPM centered around the agent's current location as input, we finetune Resnet-18 He et al. (2016) to predict a pixel coordinate directing the navigation agent's motion in the environment. To make the feudal agent goal-directed (which is necessary for the image-goal task used for evaluation), keypoint matches between the current observation and a goal image are computed by Superglue Sarlin et al. (2020). If the confidence $\alpha_k$ of this keypoint match is high, the average of the matched keypoints is used in the navigation pipeline instead of the waypoint prediction for the mid-level manager. In this manner the agent mimics human navigation in novel environments while checking if the goal location has been found. While the high-level manager reasons about the navigation task at a more global view with a coarser grained spatial scale, the mid-level manager (WayNet) has a first person, fine-grained view of the environment. Breaking the problem into these two scales introduces spatial abstraction which allows these two networks to work in tandem and compound their solutions to smaller pieces of the overall problem to perform more efficiently than an end-to-end implementations (see Section 5). This simplification also allows for faster training with smaller amounts of data (see Section 4).

**Low-Level Worker: Action**   The low level worker agent takes actions in the environment based on the current depth map and the waypoint predicted by WayNet. We define the following action space in accordance with other SOTA methods Hahn et al. (2021); Yadav et al. (2022); Wasserman et al. (2023): "turn left $15°$", "turn right $15°$", and "move forward $0.25m$". Although an RL agent is typically used for this type of task, we find an MLP classifier works well to enable effective navigation. This classifier is trained to learn a mapping between the human-chosen action from the LAVN dataset Johnson et al. (2024) and the corresponding depth map and waypoint input. The agent chooses to stop navigation when the confidence threshold $\alpha_m$ for matching goal image features to the current observation is high and either the agent's depth measurement indicates it is sufficiently close to the goal location ($\leq 1m$) or the ratio $\psi$ of the area of the matched keypoints to the total image size is relatively large.

## 4   EXPERIMENTS

**Image-Goal Navigation Task**   We test the performance of our method (FeudalNav) using the procedure outlined in NRNS Hahn et al. (2021) on the image-goal task in previously unseen Gibson Xia et al. (2018) test environments. To start, the agent is placed in an environment and given RGBD image observations of the first person view of their surroundings and a goal location. All images are $480 \times 640$ pixels with $120°$ field of view. A trial terminates if the agent stops within $1m$ of the location of the goal image or the agent takes 500 actions in the environment. Each agent trajectory is evaluated on success rate (whether or not the goal has been reached) and SPL (success weighted by inverse path length), defined as

$$SPL = \frac{1}{N} \sum_{i=0}^{N} S_i \frac{l_i}{\max(l_i, p_i)} \tag{1}$$

where $N$ is the total number of trajectories considered, $S_i$ is an indicator variable for success, $l_i$ is the optimal (shortest) geodesic path length between the starting location and the goal, and $p_i$ is the actual path length the agent traveled.

**Training and Testing Procedure**   We train our method with the LAVN Johnson et al. (2024) human navigation dataset, using a subset of 117 trajectories with a total of 36834 frames. The average number of contrastive learning training data clusters per trajectory is 23 (median 25). We train all models on either a GTX TITAN X or RTX 2080 Ti using a learning rate of 1e-4. The confidence thresholds $\alpha_c$, $\alpha_k$, and $\alpha_m$ are chosen empirically to be 0.7, and $\psi$ is similarly chosen as 0.85. The isomap imitator network used to make the MPM is a two layer MLP with ReLU activations trained for 2000 epochs with batch size= 32, and we train SMoG Pang et al. (2022) for 50 epochs with batch size = 16. We use the current observation and a $480 \times 640$ pixel crop of

the MPM centered around the agent's current observation as input to WayNet, which is a modified Resnet-18 He et al. (2016) that accepts 7 channel input trained for 250 epochs with batch size = 16. The classifier network for the low-level worker is a four layer MLP with PReLU activations where the depth map and waypoint input have distinct projection heads trained for 2 epochs with batch size = 128. In total, our entire method trains in 3M iterations. Compare this to other SOTA that uses RL and simulators and trains for anywhere from 10-100M iterations Wijmans et al. (2019); Al-Halah et al. (2022); Hahn et al. (2021) (or even 50 GPU days Yadav et al. (2022)) using anywhere from 14.5-500M frames on up to 64 GPUs. We use several orders of magnitude less data, significantly less GPUs, and a fraction of the total iterations to train our network compared to other SOTA.

We test our network using the testing procedure and baselines outlined in NRNS Hahn et al. (2021). Testing trajectories come from a publicly available set of Gibson Xia et al. (2018) environments listed in Hahn et al. (2021). They consist of approximately $6K$ point pairs (start and goal locations) that are uniformly sampled from fourteen environments and divided between two curvatures (straight/curved) and three goal distances ($1.5 - 3m, 3 - 5m, 5 - 10m$). We compare our method performance against flat RL DD-PPO Wijmans et al. (2019) trained for varying lengths of time , behavior cloning (BC) with a resnet-18 backbone and either a GRU or a metric map from Hahn et al. (2021), NRNS Hahn et al. (2021) with and without noise, ZSEL Al-Halah et al. (2022), OVRL Yadav et al. (2022), and NRNS and OVRL enhaced by SLING Wasserman et al. (2023). These methods are either trained directly in a simulator or for extended periods of time (ie. 100M time steps, 53 GPU days), require odometry, or use graphs in their implementations. There are other recent works that test on the image-goal task that require testing in previously seen environments, full scene reconstruction Kwon et al. (2023), or full panoramic or semantic images Kim et al. (2023) that are very reliant of the fixed spacing or semantic contextual information of residential dwellings. These are excluded as unfair comparisons since our method does not assume prior knowledge of semantic context (since it is not readily available in many applications).

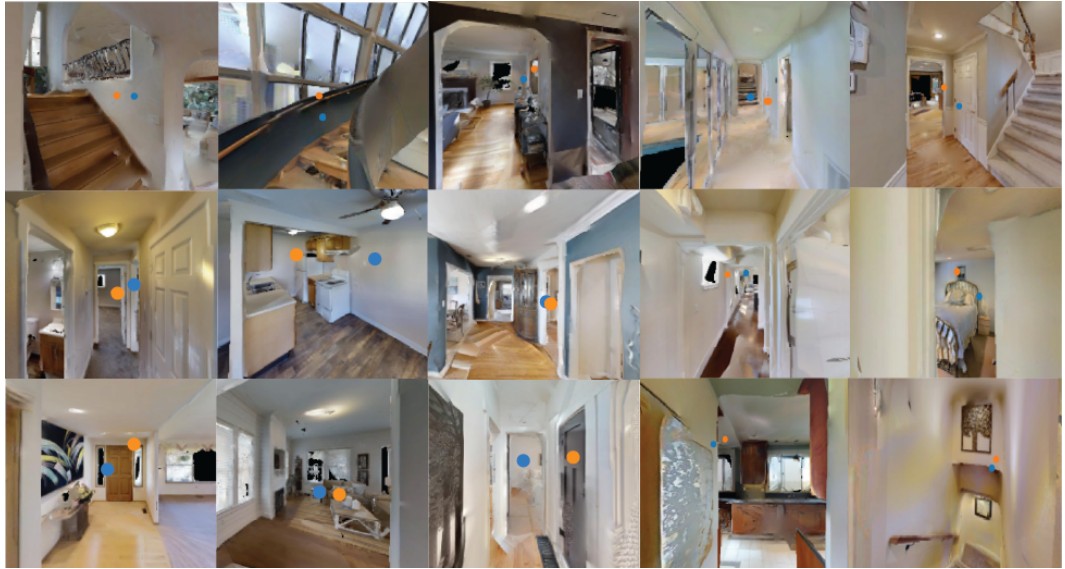

Figure 3: (Best viewed zoomed) We show qualitative results for the waypoints predicted by WayNet (blue) shown with the ground truth human click points from the LAVN dataset Johnson et al. (2024) (orange). Note that the majority of the samples show high overlap between the two. When they diverge, the WayNet waypoints still lead to navigably feasible areas in each observation, showing that our network sufficiently learns an acceptable navigation policy.

## 5 RESULTS

**Mid-Level Manager Accurately Mimics Human Navigation**  Figure 3 shows qualitative examples of the mid-level manager's predicted waypoints (blue) shown with the human ground truth point clicks (orange) from the LAVN dataset Johnson et al. (2024). For the majority of the points, both the predictions and the ground truth lie in the same area or on the same object. When this is not the case,

| | Model | Easy | | Medium | | Hard | | Average | |
|---|---|---|---|---|---|---|---|---|---|
| | | Succ↑ | SPL↑ | Succ↑ | SPL↑ | Succ↑ | SPL↑ | Succ↑ | SPL↑ |
| **STRAIGHT** | DDPPO (10M steps) * | 10.5 | 6.7 | 18.1 | 16.2 | 11.8 | 10.9 | 13.5 | 11.2 |
| | DDPPO (extra data + 50M steps) * | 36.3 | 34.9 | 35.7 | 34.0 | 6.0 | 6.3 | 26.0 | 25.1 |
| | DDPPO (extra data+100M steps) * | 43.2 | 38.5 | 36.4 | 35.0 | 7.4 | 7.2 | 29.0 | 26.9 |
| | BC w/ ResNet + Metric Map | 24.8 | 24.0 | 11.5 | 11.3 | 1.4 | 1.3 | 12.6 | 12.2 |
| | BC w/ ResNet + GRU | 34.9 | 33.4 | 17.6 | 17.1 | 6.1 | 5.9 | 19.5 | 18.8 |
| | NRNS w/ noise | 64.1 | 55.4 | 47.9 | 39.5 | 25.2 | 18.1 | 45.7 | 37.7 |
| | NRNS w/out noise | 68.0 | 61.6 | 49.1 | 44.6 | 23.8 | 18.3 | 47.0 | 41.5 |
| | NRNS + SLING | **85.3** | 74.4 | 66.8 | 49.3 | 41.1 | 28.8 | 64.4 | 50.8 |
| | OVRL + SLING * | 71.2 | 54.1 | 60.3 | 44.4 | 43.0 | 29.1 | 58.2 | 42.5 |
| | **FeudalNav (Ours)** | 82.6 | **75.0** | **71.0** | **57.4** | **49.0** | **34.2** | **67.5** | **55.5** |
| **CURVED** | DDPPO (10M steps) * | 7.9 | 3.3 | 9.5 | 7.1 | 5.5 | 4.7 | 7.6 | 5.0 |
| | DDPPO (extra data + 50M steps)* | 18.1 | 15.4 | 16.3 | 14.5 | 2.6 | 2.2 | 12.3 | 10.7 |
| | DDPPO (extra data+100M steps)* | 22.2 | 16.5 | 20.7 | 18.5 | 4.2 | 3.7 | 15.7 | 12.9 |
| | BC w/ ResNet + Metric Map | 3.1 | 2.5 | 0.8 | 0.7 | 0.2 | 0.2 | 1.4 | 1.1 |
| | BC w/ ResNet + GRU | 3.6 | 2.9 | 1.1 | 0.9 | 0.5 | 0.4 | 1.7 | 1.4 |
| | NRNS w/ noise | 27.3 | 10.6 | 23.1 | 10.4 | 10.5 | 5.6 | 20.3 | 8.8 |
| | NRNS w/out noise | 35.5 | 18.4 | 23.9 | 12.1 | 12.5 | 6.8 | 24.0 | 12.4 |
| | ZSEL* | 41.0 | 28.2 | 27.3 | 18.6 | 9.3 | 6.0 | 25.9 | 17.6 |
| | OVRL* (53 GPU days) | 53.6 | 31.7 | 47.6 | 30.2 | 35.6 | 21.9 | 45.6 | 28.0 |
| | NRNS + SLING | 58.6 | 16.1 | 47.6 | 16.8 | 24.9 | 10.1 | 43.7 | 14.3 |
| | OVRL + SLING * Wasserman et al. (2023) | 68.4 | 47.0 | 57.7 | 39.8 | 40.2 | **25.5** | 55.4 | 37.4 |
| | **FeudalNav (Ours)** | **72.5** | **51.3** | **64.4** | **40.7** | **43.7** | 25.3 | **60.2** | **39.1** |

Table 1: We show competitive results on the image goal task following the evaluation protocol from NRNS Hahn et al. (2021) in previously unseen Gibson environments Xia et al. (2018). The top results are bolded for each category in this quantitative comparison between our method (FeudalNav), baselines (DDPO Wijmans et al. (2019), NRNS Hahn et al. (2021)), and SOTA methods (ZSEL Al-Halah et al. (2022), OVRL Yadav et al. (2022), SLING Wasserman et al. (2023)). (* denotes the use of a simulator during training)

it is important to note that there may be multiple "correct" exploration waypoints in a given scene (ie. with a T junction or a hallway with multiple doors). When the mid-level manager's predicted waypoints diverge from the human point clicks, they still lead to feasibly navigable areas such as doorways or further into rooms or hallways, thus demonstrating that the network successfully mimics human point-click teleoperation and learns a useful policy for exploration.

**FeudalNav Outperforms Baselines and SOTA**   We show performance on the image-goal task as detailed in section 4 for our feudal navigation agent (FeudalNav) and SOTA in Table 1. We also report performance averaged across the easy, medium, and hard trials, which we use to conduct comparisons of model performance. Our method has shown a significant improvement in success rate performance of 108% (straight) and 283% (curved) over all DDPO Wijmans et al. (2019) baselines and 208% (straight) and 3380% (curved) over both behavior cloning (BC) methods Hahn et al. (2021) while using no RL, learning no metric map, and not training directly in a simulator. We achieve a 5% increase in performance on average success rate over NRNS+SLING Wasserman et al. (2023) and a 16% increase over OVRL+SLING Wasserman et al. (2023) on straight trajectories. We make larger improvements to SPL performance as well (9% and 31% over NRNS+SLING and OVRL+SLING respectively), despite not using odometry, not explicitly parsing semantics, not utilizing a graph, and only using $\sim 37K$ images for training (compared to the 3.5 million used by NRNS and 14.5 million used by OVRL). This trend of performance improvement continues for the curved case where FeudalNav has a 9% increase on average success rate over NRNS+SLING and 38% over OVRL+SLING. This corresponds with a 173% and 5% increase in SPL against these two methods respectively. In the real world, it is less likely that a robot will be tasked to find an object



Figure 4: Distance matrices showing a heatmap of metric distances between each pair of images in a single trajectory (450 image sequence). (Left) Ground truth distance matrix (brighter is farther away) of the locations of each image in the simulated environment. Compare this to each of the image feature-distance matrices (computed using MSE) for the same images pairs using features from the following (from left to right after GT distance): SMoG Pang et al. (2022), MocoHe et al. (2020), Resnet He et al. (2016), and Swav Caron et al. (2020). The more the feature distances resemble the ground truth metric distances, the more effective the features will be at encoding a proxy for relative distance between real world observations. We choose SMoG for feature detection in our high-level manager because its feature distance matrix most closely resembles the ground truth distances heatmap.

within a straight line of sight from itself. For this reason, performance on the curved trajectory case gives a more realistic prediction of real world performance.

## 6  ABLATION STUDY

**Different Network's Effects on the Memory Proxy Map**    We test the effectiveness of using multiple real world views as a contrastive learning augmentation. In order for this to occur, we postulate that the image feature distances between two points must be highly correlated with the ground truth distances between those points. In Figure 4, the ground truth distances between pairs of observations for a single trajectory from the LAVN dataset Johnson et al. (2024) using the Copemish Gibson environment are compared to the distance between predicted image features for our high level manager network, which utilizes SMoG Pang et al. (2022). For baselines, we also compare the inter-observation feature distance matrices for Moco He et al. (2020), Resnet-18 He et al. (2016), and SwavCaron et al. (2020). In the figure, lower distances are darker/purple and higher distances are lighter/yellow. The more a latent space mimics geometric distances between image features, the more it should resemble the first square showing the distance matrix created using ground truth distances.

Swav Caron et al. (2020) learns highly diverse features (large distances between the learned features), while Resnet He et al. (2016) has the reciprocal issue (highly similar features for all of the images in the trajectory) as shown in Figure 4. This is understandable because the diversity between separate views of an indoor scene is relatively small compared to the diversity of samples used to train Resnet. However, this means that both of these methods' features are unsuitable for navigation because they do not provide a useful distance proxy. Moco He et al. (2020) begins to mimic the ground truth distance matrix, but still learns features that are largely similar to each other. Validating our intuition, SMoG Pang et al. (2022) learns the most distance preserving latent space due to its ability to optimize inter-sample and inter-cluster distances simultaneously, making it the most useful for navigation purposes.

**Full Hierarchy's Effect on Image-Goal Task Performance**    We also provide an ablation study to show how important each level of hierarchy is to our overall results in Table 2. We incrementally add each piece of our architecture together and report the image-goal task results for the same experiment listed in Section 4. In the table, the second column indicates whether or not the high level manager's memory proxy map (MPM) is included in the feudal navigation agent. Networks without this module are denoted by –. $C$ denotes a binary representation of the map where $M(x, y) = 1$ for a circular region around previously seen *cluster centers* where the radius ($r$) of the circular region grows linearly with respect to the frequency of visitation. $A$ denotes the binary map where $r$ is fixed and all observations are added to the map, and $H$ denotes the MPM as detailed in Section 3. For the mid-level manager, we test versions of WayNet that take in a single RGB, RGBD, or RGBD-MPM

| | MPM | WayNet | LLW | Easy | | Medium | | Hard | | Average | |
|---|---|---|---|---|---|---|---|---|---|---|---|
| | | | | Succ↑ | SPL↑ | Succ↑ | SPL↑ | Succ↑ | SPL↑ | Succ↑ | SPL↑ |
| **STRAIGHT** | - | RGB | Det | 48.00 | 30.28 | 37.00 | 21.75 | 24.57 | 13.21 | 36.52 | 21.75 |
| | - | RGBD | Det | 48.20 | 31.70 | 39.40 | 21.50 | 24.94 | 12.99 | 37.51 | 22.06 |
| | - | 3 RGBD | Det | 50.20 | 31.19 | 38.60 | 21.24 | 20.72 | 9.16 | 36.51 | 20.53 |
| | C | RGBD-M | Det | 61.40 | 51.81 | 50.30 | 40.10 | 31.02 | 23.33 | 45.73 | 37.69 |
| | C | 3 RGBD-M | Det | 65.90 | 48.50 | 51.00 | 29.22 | 33.62 | 14.49 | 50.17 | 30.74 |
| | A | RGBD-M | Det | 57.90 | 50.35 | 44.80 | 37.70 | 30.52 | 24.98 | 44.41 | 37.68 |
| | A | 3 RGBD-M | Det | 65.90 | 49.26 | 51.00 | 29.98 | 27.17 | 13.75 | 48.02 | 31.00 |
| | H | RGBD-M | Det | 72.00 | 64.55 | 60.40 | 50.96 | 41.32 | 34.01 | 57.91 | 49.84 |
| | H | 3 RGBD-M | Det | 77.5 | 64.53 | 62.50 | 42.18 | 44.29 | 24.50 | 61.43 | 43.74 |
| | **H** | **RGBD-M** | **Cl** | **82.60** | **74.95** | **71.00** | **57.40** | **49.01** | **34.20** | **67.54** | **55.52** |
| | H | 3 RGBD-M | Cl | 73.60 | 73.05 | 37.10 | 35.66 | 9.18 | 8.95 | 39.96 | 39.22 |
| **CURVED** | - | RGB | Det | 34.70 | 11.20 | 32.60 | 13.02 | 18.20 | 7.24 | 28.5 | 10.49 |
| | - | RGBD | Det | 36.60 | 11.55 | 30.00 | 11.86 | 18.30 | 7.72 | 28.3 | 10.37 |
| | - | 3 RGBD | Det | 39.50 | 11.91 | 32.80 | 11.75 | 15.70 | 5.65 | 29.33 | 9.77 |
| | C | RGBD-M | Det | 41.30 | 19.51 | 32.60 | 17.10 | 18.60 | 10.88 | 30.83 | 15.83 |
| | C | 3 RGBD-M | Det | 56.40 | 21.37 | 44.10 | 17.41 | 21.00 | 7.30 | 40.50 | 15.36 |
| | A | RGBD-M | Det | 35.70 | 18.44 | 31.10 | 19.00 | 17.20 | 11.16 | 28.00 | 16.2 |
| | A | 3 RGBD-M | Det | 56.60 | 22.19 | 43.30 | 15.99 | 19.8 | 7.39 | 39.90 | 15.19 |
| | H | RGBD-M | Det | 53.80 | 27.91 | 42.60 | 25.00 | 27.20 | 17.01 | 41.2 | 23.31 |
| | H | 3 RGBD-M | Det | 68.60 | 28.93 | 56.40 | 26.06 | 32.40 | 14.44 | 52.47 | 23.14 |
| | **H** | **RGBD-M** | **Cl** | **72.50** | **51.26** | **64.40** | **40.73** | **43.70** | **25.32** | **60.2** | **39.11** |
| | H | 3 RGBD-M | Cl | 59.00 | 55.52 | 15.50 | 14.58 | 1.50 | 1.45 | 25.33 | 23.85 |

Table 2: An ablation study showing the effect of each module in FeudalNav on overall image-goal task performance. For the second column, – denotes a network without the MPM from the high-level manager, C denotes the use of an MPM where only the cluster centers are plotted in RGB, A denotes an MPM where all observations are plotted in RGB, and H denotes the MPM described in section 3. For Waynet, we use RGB or RGBD input with the MPM (denoted by the -M) or without the MPM, either of a single observation or of three historical observations. For the low-level worker, "Det" refers to a worker that deterministcally maps waypoint locations to actions, and "Cl" refers to the classification network specified in section 3. **The official FeudalNav implementation is bolded.** Notice the performance gains attributed to the MPM, specially gaussian weighted MPM (denoted by H).

(denoted RBGD-M in the table) input and versions that take in three historical timesteps worth of each of these respective inputs (3 RGBD and 3 RGBD-M). We test two versions of the low-level worker: one that deterministically maps waypoint image coordinate predictions to environment actions ("Det") and one that follows the classifier approach detailed in Section 3 ("Cls").

For the networks without the MPM, we found that WayNet using RGB input performed qualitatively worse than using RGBD input, despite their similar performance, because depth information allowed the agent to avoid obstacles more efficiently. There is a $25\%$ increase in performance when the MPM is added to our network. This shows that the memory component is crucial to image-goal navigation success. However, the form this memory module takes is also an important factor. We see a $28\%$ increase in performance between the gaussian heatmap ("H") and the binary heat maps ("C" and "A"). Furthermore, we find that a learning-based approach performs best for obstacle avoidance with a $10\%$ increase from the deterministic ("Det") to the classification ("Cls") low-level worker.

## 7 CONCLUSION

Our work extends prior research on visual navigation for the image goal task by providing a high performance, no-RL, no-graph, no-odometry solution.[1] The resulting methodology is efficient, lightweight and demonstrably effective. Our frameworks accomplishes this by putting emphasis on building representations for *agent memory*, i.e. the memory proxy map (MPM). Such an emphasis is critically important for no-graph solutions and can be extended for new paradigms of visual navigation that include continual learning. With the ubiquity of advanced SLAM algorithms, the questions

---

[1]Code released upon publication.

arise: If SLAM is solved, why learn?, and why use visual navigation instead of building a metric map? Certainly, there are many applications where SLAM is the best solution. But, SLAM is typically computationally intensive and requires considerable development time and compute resources since each new environment requires management of a large-scale optimization problem. Visual navigation holds the promise of a lightweight solution, suitable for select applications. Additionally, visual and learning-based navigation enables systems to learn about the dynamic environments agents are charged with navigating, including learning the nuances of navigating in social spaces. Visual navigation also holds the promise of a low-cost, low-latency solution suitable for small mobile agents at an accessible price point. In the long term, visual navigation applications with positive societal impact can enable intelligent navigation agents for applications in unseen environments such as disaster assessment, environment monitoring in hazardous areas (volcanic, subterranean), and indoor navigation of previously unmapped buildings.

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
