# OpenReview forum: "Memory Proxy Maps for Visual Navigation"
_ICLR.cc/2025/Conference — Submitted to ICLR 2025_

### Official Review · Reviewer_7YiN · 2024-10-28

**Soundness:** 3
**Presentation:** 3
**Contribution:** 2
**Rating:** 5
**Confidence:** 4

**Summary:**

The authors introduce a hierarchical system for visual navigation that stores embeddings of intermediate subgoals and their connectivity. They propose a three-tiered hierarchical scheme where the highest level is responsible for maintaining a map of semantically similar states, the middle level determines medium-scale navigation instruction akin to human point-and-click navigation, and the lowest level converts point-and-click commands into motion actions. These components are trained with an unsupervised clustering/contrastive scheme for the high-level embeddings and supervised learning on human demonstrations for the mid/low-level policies. The scheme is empirically validated in Habitat environments, yielding SOTA performance.

**Strengths:**

- The method seems relatively simple to implement, using mostly off-the-shelf losses and network components
 - The method yields strong performance on the held-out Gibson navigation environments.

**Weaknesses:**

Method:
 - The method assumes a (nonlinear) projection from the latent space to the environment, which seems questionable: an environment may certainly have multiple locations with similar or near-identical features that are differentiated only by connectivity with other locations in the graph.
 - The ablations, while a thorough hyperparameter sweep, do not make it clear exactly what components of the proposed system impact performance, or why. It would be nice to ablate components in the hierarchical scheme directly (e.g. what if the mid/low level are combined into a single policy) as well as maybe demonstrating whether the SMoG loss (vs. simple contrastive learning) is actually necessary.
    - The novelty of this paper mostly lies in the composition of these components into a system that works well, so it is important to determine which parts of the system are actually important.

Presentation:
 - In general, method description could be improved (in particular, being more concrete early in the paper about how the MPM is represented/built)
 - In general, the focus on calling the method "feudal learning" seems a little misplaced, given that the term originally refers to a specific system in RL (which is not used here), and just referring to "hierarchical learning" would probably be more widely understood.

**Questions:**

Clarifying questions:
 1. Is the isomap imitator network learned online from scratch for any new environment?
 2. Regarding claims at the end of the second paragraph of the introduction: I see why not requiring RL or simulator-based training are desirable, but why the desiderata for "no odometry", "no graphs", and "no metric maps"? In some sense doesn't the model build it's own graph/map with the MPM?
 - It seems like the MPM is represented to the mid-level agent as an image - why use this rather than e.g. attention over previous points?

---

### Official Review · Reviewer_3oN5 · 2024-10-30

**Soundness:** 3
**Presentation:** 2
**Contribution:** 3
**Rating:** 6
**Confidence:** 4

**Summary:**

The paper presents a novel approach to visual navigation with no-RL, no-graph, and no-odometry.  Inspired by how humans navigate unfamiliar environments using vision, the authors propose a three-tiered feudal learning system that comprises a high-level manager, a mid-level waypoint generator, and a low-level worker agent. Key to this system is the Memory Proxy Map (MPM), which provides a lightweight, self-supervised representation of the environment as a proxy for memory. The mid-level waypoint network (WayNet) generates intermediate subgoals for local navigation, mimicking human behavior, while the low-level worker handles obstacle avoidance and goal-directed movement. The proposed method achieves the SOTA result on the image goal navigation task.

**Strengths:**

(1) The paper proposes a hierarchical and pure vision-based navigation framework. The proposed method achieves the best performance on the ImageNav benchmark without RL, explicit map, and odometry. This mapless navigation framework significantly reduces computational complexity for robot applications.
(2) The proposed idea of using dimension-reduced feature space as a map (Memory Proxy Maps) is novel and lightweight compared with metric-based maps and topological graphs.
(3) The experiment performance shows the proposed method's superiority over the SOTA ImageNav baselines, even without any intricate graph and odometry.
(4) The training process is much more efficient than the RL-based navigation methods, which can potentially be further improved with a more diverse dataset.

**Weaknesses:**

(1) The entire system is complicated with 3 layers of decision modules, the lack of necessary ablation studies (especially lacking some quantitive metrics) makes it difficult to comprehensively understand the functions of some proposed components. For example, Figure 3 provides the visualization of the difference WayNet prediction coordinates and the ground-truth coordinates. What is the average distance between the prediction and the labels across diverse scenes? Quantitive metrics should be reported to help understand the generalization performance of the WayNet. The same for Figure 4, it is better to report the navigation performance with the MPM built with different features, instead of only showing the similarity visualization.

(2) Although a mapless approach for navigation is favorable and appealing, the sim-to-real gap in vision is a challenging problem for a real navigation robot system. Can your Memory Proxy Map generalize well with real-world indoor images as input? A cluster visualization like Figure 2 with real-world captured image trajectory can help highlight the contribution of MPM.

(3) It is unclear whether the Memory Proxy Map can be effectively utilized for other navigation tasks, such as object navigation[1] and navigation exploration[2].

(4) Some closely related works can be added. Such as PixNav[3] which selects a pixel goal with LLMs and trains a local action predictor to navigate to the pixel goal. And the citation of isomap should also be appended.

[1] Chaplot, Devendra Singh, Dhiraj Gandhi, Abhinav Kumar Gupta and Ruslan Salakhutdinov. “Object Goal Navigation using Goal-Oriented Semantic Exploration.” ArXiv abs/2007.00643 (2020): n. pag.

[2] Chaplot, Devendra Singh, Dhiraj Gandhi, Saurabh Gupta, Abhinav Kumar Gupta and Ruslan Salakhutdinov. “Learning to Explore using Active Neural SLAM.” ArXiv abs/2004.05155 (2020): n. pag.

[3] Cai, Wenzhe, Siyuan Huang, Guangran Cheng, Yuxing Long, Peng Gao, Changyin Sun and Hao Dong. “Bridging Zero-shot Object Navigation and Foundation Models through Pixel-Guided Navigation Skill.” 2024 IEEE International Conference on Robotics and Automation (ICRA) (2023): 5228-5234.

**Questions:**

(1) It seems that the positive and negative pairs for representation learning are selected by considering the visual similarity by key points matching. But in the navigation tasks, the agent can make rotations, which can cause the robot to stand at the same position but with significant differences in observation images. Why not use the temporal distance-based rules to select instead of the visual distance-based rules? Does the use of visual distance-based representation learning cluster the observation images with only different yaw angles into different coordinates in MPM?

(2) The paper frequently mentions the LAVN dataset but does not clearly specify what information is included in the dataset (e.g., images, waypoints, depth maps, actions) or the environments it covers (e.g., Gibson, Matterport). It is recommended to add a brief description of the specific data types and information used from the LAVN dataset to improve the readability of the paper.

(3) In the ablation experiments regarding the low-level worker, how are the actions determined in the Det version? What kind of rules (e.g., geometric or simple logic rules) are used to decide the actions?

(4) Why just changing the RGBD input numbers cause such a severe drop in the overall performance?

---

### Official Review · Reviewer_r9uZ · 2024-10-31

**Soundness:** 3
**Presentation:** 3
**Contribution:** 2
**Rating:** 5
**Confidence:** 4

**Summary:**

The paper proposes a new method for image goal conditioned visual navigation in indoor environments with no-RL training, without explicit graphs of the environment, and no odometry. The key idea in proposed approach approach is to build a modular method that  - (1.) First builds a latent memory representation of the environment called a memory proxy map (MPM) using a self-supervised approach, (2.) Next, use a learned waypoint network (WayNet) that takes as input current observation and snapshot of latent memory representation and outputs a intermediate subgoal (waypoint in image space) to navigate to, (3.) Lastly, a low-level navigation agent takes as input current observation and waypoint in image space and outputs discrete navigation actions to navigate to predicted waypoint. Using this feudal learning based method authors claim that they can achieve state-of-the-art results on ImageGoal navigation task in Gibson dataset without using RL, explicitly graph of environment and odometry.

**Strengths:**

1. The paper is easy to follow and the key ideas are explained clearly
2. The idea of breaking down the task into separate task-specific components has benefits like interpretability and improvement in sample efficiency
3. The idea of building a latent memory representation using self-supervised learning is novel and a interesting contribution.
4. Similarly, the idea of training a waypoint network using LAVN dataset is promising as it enables possibilities for training navigation agents without requiring simulation and annotated robot actions.

**Weaknesses:**

1. One main concern I have with the paper is that the authors claim the method is SoTA on the ImageNav while it is not. A end-to-end method presented in [1] last year achieved significant improvements in ImageNav and InstanceImageNav performance. Comparison with this method is missing in the paper. I would appreciate if authors can add a comparison or explain why comparison with this method wouldn’t be fair and update the claim in the paper accordingly. Additionally, comparison to another method OVRL-v2 [2] is also missing in the paper.
2. In second section of section 3, the description of how the memory proxy map is used as input for waypoint network is not very well explained which makes it quite hard to understand the method. Can authors please clarify how the latent representation snapshot is fed to the WayNet network?

[1] End-to-End (Instance)-Image Goal Navigation through Correspondence as an Emergent Phenomenon

[2] OVRL-V2: A simple state-of-art baseline for ImageNav and ObjectNav

**Questions:**

Additional suggestions:

1. The table used inablation study section can be improved. In its current state it is quite hard to understand the changes being made. I would recommend authors to add separate column for each component they are ablating and use 1-2 word to describe that component in heading and use tick or cross marks to indicate whether it is enabled of disabled. The results on each gibson split can be reported as aggregate numbers for success and SPL.

---

### Official Review · Reviewer_WUhh · 2024-11-02

**Soundness:** 2
**Presentation:** 1
**Contribution:** 2
**Rating:** 3
**Confidence:** 4

**Summary:**

The paper introduces a novel approach to visual navigation that does not rely on reinforcement learning, graphs, odometry, or metric maps. The key components are a memory proxy map (MPM) learned in a self-supervised manner by a high-level manager agent, a waypoint network (WayNet) that outputs intermediate subgoals for a mid-level manager agent to imitate human navigation, and a low-level worker agent that avoids obstacles and moves towards the waypoints. This feudal learning framework allows the agents to operate at different spatial and temporal scales to tackle the visual navigation task in previously unseen environments. The authors demonstrate state-of-the-art performance on the image-goal navigation task in the Gibson simulation environment without requiring any odometry information or training in simulators.

**Strengths:**

1. The paper proposes a novel modular navigation method that does not rely on reinforcement learning, odometry, or metric maps. The memory proxy map is learned through the self-supervised SMoG method, which does not require human supervision or environmental odometry.
2. Extensive experiments on the Gibson dataset have demonstrated the SOTA performance of FeudalNav, compared with RL-based and graph-based baselines. The ablation study also verifies the effectiveness of modules (especially the MPM map) in FeudalNav.

**Weaknesses:**

The paper is poorly written and fails to deliver a convincing motivation or clear methodology.
1. The whole hierarchical method is not described formally, which makes multiple technical details unclear, including the momentum grouping model (line 158), visual similarity (line 161), confidence of keypoint matches (line 168), gaussian window (line 180), WayNet (line 229), and so on.
2. The approach of imitating human waypoint generation through supervised learning is questionable. First, the statement that it's "generalizable to new environments with zero-shot transfer" (line 68) is not verified, since there's a high-level MLP and low-level classifier trained in the Gibson environment and the LAVN dataset also contains images sampled from the Gibson dataset. Experiments in new environments (e.g. MP3D, AI2THOR, or the real world) should be done, otherwise, collecting human datasets for supervised training can be more costly than RL-based or map-based approaches. Second, the intuition in line 65 is unreasonable. Why do humans tend to choose a waypoint to navigate instead of other ways, such as choosing a direction with a joystick? Third, the qualitative results in Figure 3 show samples in separate episodes which is difficult to prove how well the model imitates human strategy. It's better to showcase several sequential keyframes in an episode and analyze why such decisions are effective and transferable.
3. It's not explained why taking a single observation is better than three historical observations when using a low-level classifier, while it's obviously better to take three observations with the deterministic worker (Table 2).

**Questions:**

1. How does the MPM allow the agent to localize itself (line 184)? The proxy map does not seem to provide any temporal information.
2. Explain the method and the questions mentioned in the weaknesses section.

---

### Meta-Review · Area_Chair_rmKL · 2024-12-11

**Metareview:**

This paper introduces a hierarchical approach to visual navigation that eliminates reliance on reinforcement learning, graphs, odometry, or explicit metric maps. The method proposes a three-tiered feudal learning system, including a high-level manager using a self-supervised Memory Proxy Map (MPM) for environment representation, a mid-level waypoint network (WayNet) for generating intermediate subgoals, and a low-level worker agent responsible for avoiding obstacles and moving towards waypoints.

The simple modular method, motivation of developing latent memory representation using self-supervised learning, and results on held-out test Gibson scenes was appreciated by the reviewers. Reviewers raised concerns about clarity in explaining some of the key technical components, such as the Memory Proxy Map (MPM), WayNet, and momentum grouping, mentioning that these details weren’t fully clear. They also questioned the zero-shot generalization claims since the experiments were limited to the Gibson dataset and didn’t include unseen domains, eg. MP3D, AI2THOR, or real-world settings. Concerns were raised about missing comparisons to end-to-end ImageNav and OVRL-v2 (published state-of-the-art in ImageNav). Some reviewers mentioned missing ablations and details. The term "feudal learning" was also flagged as a bit misleading.

The AC concurs with the majority of reviewers and notes that the concerns raised remain unresolved due to no response from the authors. The detailed feedback from reviewers highlights several areas that need clarification and improvement, which could be addressed in future revisions for a potential resubmission.

**Additional Comments On Reviewer Discussion:**

No rebuttal was submitted by the authors, and no further discussion occurred among the reviewers.

---

### Decision · Program_Chairs · 2025-01-22

Reject